# Cancer Treatment and Immunotherapy during Pregnancy

**DOI:** 10.3390/pharmaceutics14102080

**Published:** 2022-09-29

**Authors:** Antonios Koutras, Thomas Ntounis, Zacharias Fasoulakis, Theodoros Papalios, Savia Pittokopitou, Ioannis Prokopakis, Athanasios Syllaios, Asimina Valsamaki, Athanasios Chionis, Panagiotis Symeonidis, Athina A. Samara, Athanasios Pagkalos, Vasilios Pergialiotis, Marianna Theodora, Panos Antsaklis, Georgios Daskalakis, Emmanuel N. Kontomanolis

**Affiliations:** 11st Department of Obstetrics and Gynecology, General Hospital of Athens ‘ALEXANDRA’, National and Kapodistrian University of Athens, Lourou and Vasilissis Sofias Ave, 11528 Athens, Greece; 21st Department of Surgery, Laikon General Hospital, National and Kapodistrian University of Athens, Agiou Thoma Str. 17, 11527 Athens, Greece; 3Department of Internal Medicine, Koutlimbaneio and Triantafylleio General Hospital of Larissa, Tsakalof Str. 1, 41221 Larisa, Greece; 4Department of Obstetrics and Gynecology, Laikon General Hospital of Athens, Agiou Thoma Str. 17, 11527 Athens, Greece; 5Department of Obstetrics and Gynecology, Democritus University of Thrace, 6th km Alexandroupolis–Makris, 68100 Alexandroupolis, Greece; 6Department of Embryology, University General Hospital of Larissa, Mezourlo, 41110 Larissa, Greece; 7Department of Obstetrics and Gynecology, General Hospital of Xanthi, Neapoli, 67100 Xanthi, Greece

**Keywords:** immunotherapy, cancer, pregnancy, gestation, chemotherapy during pregnancy

## Abstract

Background/aim: Immunotherapy has, in recent years, witnessed an expansion in its indications for the treatment of cancer. Coupled with the fact that, nowadays, even more women choose to postpone parenthood, thus increasing their chances of having some kind of malignancy during pregnancy, more and more women are eligible for receiving immunotherapy during this period of their lives. The cases of cancer diagnosed during pregnancy is an ever-increasing trend nowadays. Materials and methods: The oncologists and clinicians treating women often face a range of ethical and therapeutic dilemmas due to the particularity of the patient’s conditions. The primary concern is the protection of the mother, firstly, and then the fetus (through adjustments to the various treatment regimens) if possible. Results and conclusions: Oncological drugs, radiation therapy, surgery, or a combination of all the above methods are selected, depending on the case. In this project, we studied the oncology drugs used for various types of gestational cancer, their appropriateness and timing, as well as their possible effects on the parent and embryo upon their administration. Various studies have shown that the administration of oncological drugs should be postponed until at least after the first trimester of pregnancy.

## 1. Introduction

The American Society of the Clinical Oncology Agency (ASCO) considers it safe for the outcome of pregnancy to avoid the administration of oncological preparations for specific periods. For example, they recommend avoiding oncology treatment during the first trimester of pregnancy as much as possible, as irreversible damage to the fetus or miscarriage may be caused. Often, physicians delay the administration of pharmaceutical anticancer treatment until the second or the third trimester, while others wait until after delivery. There are many cases where the administration of oncological preparations during pregnancy has led to its termination or damaged the health of the fetus or infant. Furthermore, ASCO notes that pregnancy status does not affect the outcome of anticancer drug treatment, but delaying its onset may adversely affect its outcome. Thus, the so-called ethical dilemmas about starting the administration of oncological drugs for pregnant women have been raised [1]. In any case, it is recommended to estimate the cost-benefit relationship from the administration of these drugs, with the primary concern being the health of the pregnant woman and then the well-being of the fetus [2]. Finally, women who already suffer from cancer and follow an anticancer therapeutic scheme, should be strongly advised and informed about the different contraception methods, as well as their duration which should last for up to 6 months at least after the completion of the anticancer therapy [3]. Immunotherapy is of vital importance and needs to be investigated for its safety and efficacy. The balance between protection against infections and not rejecting the semi-allogenic fetus that the immune system of the mother tries to maintain, combined with little knowledge and experience regarding the safety of this kind of medication for the fetus and the neonate, shows that it is of vital importance to investigate whether immune-modifying agents could lead to unfavorable complications during pregnancy or lactation.

The American Cancer Society defines cancer as a disease characterized by abnormal growth of the body’s cells, which end up competing with the healthy cells of the body while having the ability to infect any part of the body. Thus, different types of cancer are created, such as breast cancer, lung cancer, pancreatic cancer, and other cancers. The causes behind cancer are plenty (heredity, unhealthy lifestyle, radiation exposure, viral infections, and others), and the manifestations depend on the type and stage of cancer (four stages). The treatment approach concerning oncology drugs is based on a case-by-case basis. The appropriate management of anticancer pharmaceutical compositions may diminish or retard the growth of tumor cells. Chemotherapy, hormones or other drugs, radiotherapy (treatment with the use of special radiation), surgery, and the combination of the above methods constitute some of the choices. [1].

Although the coexistence of pregnancy and cancer did not seem to be a very common phenomenon until recently (one in a thousand pregnancies), in recent years it has shown an increasing trend and is beginning to occupy the scientific community. Modern rhythms of lifestyle, which have become more and more demanding, force women to postpone being pregnant. In that way, the possibility of developing some form of cancer before (or sometimes during) pregnancy is increased. In fact, a concept known as gestational cancer has been developed and is the definition of cancer diagnosis during pregnancy or up to twelve months after birth [2,3,4]. According to McCormik and Peterson (2018), the most common cancers of reproductive age in women are melanoma, breast cancer (the most common gestational cancer and reaches 20% of cases), thyroid cancer, cervical cancer, and lymphomas (most commonly Hodgkin’s lymphoma) [2]. A pregnancy that coexists with cancer is not an ordinary pregnancy and consists of a complex medical condition. In the majority of these cases, various therapeutic and ethical dilemmas arise [3].

The main guideline is to ensure pregnant womens’ sustainability. Additionally, protecting the fetus through adjustments to the various treatment regimens is critical. Terminating pregnancy or the induction of premature birth due to a maternal cancer diagnosis is not preferred. Instead of that, a therapeutic approach strategy is designed resulting from the collaboration of various specialty health professionals, such as the oncologist doctor, gynecologist-obstetrician specializing in oncology, and others [5,6,7]. The choice of appropriate treatment depends on the stage of the pregnancy, and the stage and the type of cancer [6,8].

The safest therapeutic approach for the fetus is considered to be an operation, where the cancerous tumors and the affected areas are excised. This is indicated in all trimesters of pregnancy. The usage of radiotherapy in pregnant women is unclear and is selected on a case-by-case basis. Regarding the administration of anti-cancer–oncological drugs, great care and careful planning is needed [9,10,11].

## 2. Pregnancy and Oncological Medicines

Thanks to modern studies, various treatment options have been created, depending on the location of each cancer (Table 1).

### 2.1. Pregnancy and Anticancer Drug Treatment for the Breast Cancer

There are four categories of oncology drugs used to treat breast cancer (Figure 1), such as [12,13]:

#### 2.1.1. The Adjuvant Oncology Drugs

These are preparations that aim to restore the body’s normal immune defenses.

Peccatori and colleagues (2009) studied the safety of administering an antineoplastic–immunomodulatory factor for treating breast cancer in pregnant women. This agent is widely known as epirubicin and proved to be quite safe and effective during pregnancy, without exhibiting a significant proportion of embryotoxicity. The initiation of the treatment was made after the first trimester of pregnancy. The dosage of epirubicin administered to women with breast cancer was 35 mg/m weekly [14]. Hahn and colleagues (2006) studied the safety of administrating a chemotherapy treatment for breast cancer, known as FAC chemotherapy regiment (Fluorouracil, Adriamycin, and Cytoxan), during pregnancy. The results indicated that administering this therapy is safe for both the mother and the fetus during the second and third trimester of the pregnancy [15]. AC (doxorubicin, cyclophosphamide) and EC (epirubicin, cyclophosphamide) were proven to be equally safe [13].

Another widely used chemotherapy for breast cancer is CMF (cyclophosphamide, methotrexate, 5-fluorouracil), contraindicated strictly in the first trimester of pregnancy as it is associated with irreversible damage to the fetus and in some cases with death. However, scientists do not prohibit the administration (with simultaneous observance of strict safety measures) during the second and the third trimester [13,16].

#### 2.1.2. Hormonal Oncology Drugs

The administration of hormonal oncological drugs is realized mainly after a mastectomy or in the advanced stages of the disease. The hormonal agents that are administered in such cases are tamoxifen, anastrozole, letrozole, and exemestane. The ASCO recommends delaying the administration of hormonal therapy after childbirth, as damage to the fetus may occur [17]. More specifically, tamoxifen administration, especially during the first three months of pregnancy, has been closely associated with the development of neonatal congenital malformations [18]. Anastrozole was associated with the phenomenon of spontaneous abortion, late fetal development, and adverse malformations of the fetal genital organs [19]. Letrozole has been blamed for high embryotoxicity and teratogenicity [20], while exemestane appears to be responsible for both fetal and maternal deaths [21].

#### 2.1.3. Oncology Drugs against HER-2 Type Breast Cancer

The most common therapeutic approach for women with HER-2 type breast cancer is the administration of the chemotherapeutic form, with lapatinib and/or trastuzumab for one year [22]. Many studies have shown that it is safe to administer this treatment during pregnancy. However, there are other studies that accuse it of spontaneous abortions, while it is considered that this field needs further investigation [23,24].

#### 2.1.4. Oncology Drugs Related to Metastatic Breast Cancer

A rare situation, with little scientific data, is metastatic breast cancer during pregnancy. Azim and Peccatori (2008) describe the relatively safe administration of anthracyclines, taxanes, epirubicin, and vinorelbine after the first three months of gestation [25].

### 2.2. Pregnancy and Anti-Cancer Drug Treatment for Melanoma (Figure 2)

Melanoma is an aggressive form of skin cancer and constitutes one of the most common cancers in pregnant women. Treating melanoma includes surgery and is followed by drug treatment in uncontrolled cases, especially in cases of metastases [26]. The oncology drugs used to treat melanoma are (1) dabrafenib, which is strictly contraindicated during pregnancy, as it has been shown to be teratogenic [26,27] and (2) ipilimumab, which should be administered in accordance with strict protocols, as it is responsible for fetal death and premature birth [26]; (3) imatinib is responsible for a large proportion of congenital malformations [28,29]. (4) Cisplatin has been shown to be responsible for miscarriages and teratogenesis [30]. Finally, (5) dacarbazine, which is considered to be the most effective approach in the treatment of metastatic melanoma and did not appear to be responsible for fetal deaths or fetal malformations. It is usually combined with other chemotherapeutic agents, such as bleomycin, vincristine, and lomustine [31,32,33].

**Figure 2 pharmaceutics-14-02080-f002:**
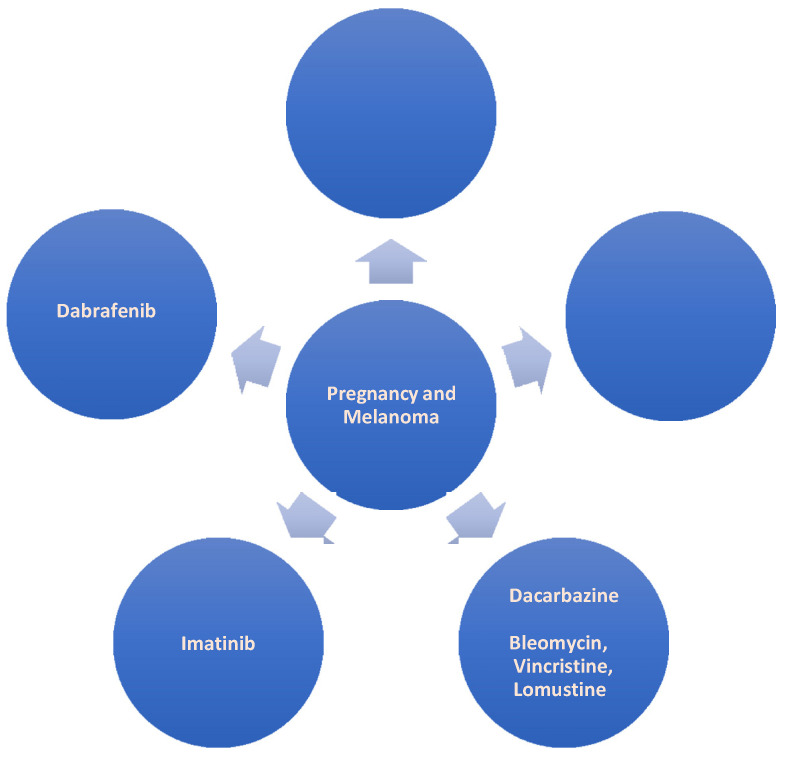
Pregnancy and Melanoma.

As a guideline, chemotherapy for melanoma should be postponed to the second or third trimester of pregnancy. The use of chemotherapeutic drugs in the first trimester has been associated with a significantly increased risk of miscarriage and congenital malformations in the newborn [31,32,33].

### 2.3. Pregnancy and Anti-Cancer Drug Treatment for the Cervix (Figure 3)

Treatment with cisplatin during pregnancy appears to be associated with low-birth-weight neonates. Additionally, administrating cisplatin and paclitaxel seems to be responsible for embryonic congenital malformations (such as hearing loss, retroperitoneal embryonal rhabdomyosarcoma, and a heterozygous mutation in the gene GJB2) [34]. However, cisplatin is the most widely used chemotherapeutic approach for cervical cancer during the second and third trimester of gestation [35]. The combination of paclitaxel–carboplatin was charged with the evolution of preterm labor, preeclampsia, intrauterine growth retardation of fetus, and transient leukopenia in the infant, when undergoing the second or third trimester of pregnancy, which makes it a less safe option [35].

**Figure 3 pharmaceutics-14-02080-f003:**
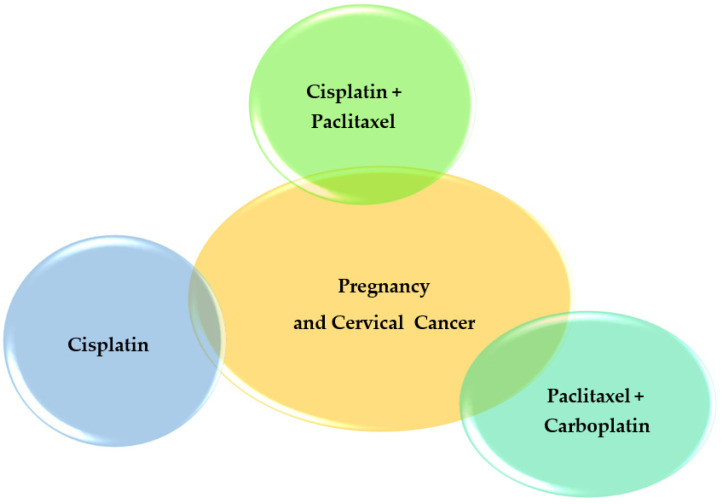
Pregnancy and Cervical Cancer.

### 2.4. Pregnancy and Anti-Cancer Drug Treatment for Ovarian Cancer (Figure 4)

Ovarian cancer is estimated to be the fifth most common form of cancer during pregnancy. The chemotherapeutic approach to the disease initiates after the first trimester of pregnancy [36]. In cases of epithelial ovarian cancer, the chemotherapeutic intervention always follows surgery [36]. The normal practice constitutes administering a combination of two types of oncological pharmaceuticals. A frequent combination involves the compound platinum (usually cisplatin or carboplatin) with another type which is called taxane, such as paclitaxel or docetaxel [37]. The most commonly used oncology drugs in the medical practice for treating epithelial ovarian cancer are paclitaxel coupled to albumin, althetamine, capecitabine, cyclophosphamide, etoposide, gemcitabine, ifosfamide, irinotecan, liposomal doxorubicin, melfalan, pemetrexed, topotecan, and vinorelbine [37].

**Figure 4 pharmaceutics-14-02080-f004:**

Pregnancy and Ovarian Cancer.

If this type of cancer is diagnosed during five weeks of gestation after an exploratory laparotomy, chemotherapy paclitaxel and carboplatin can be administered to the patient after the 17th week of gestation with relatively safety, with no apparent side effects to both the pregnant woman and the fetus [38]. Many studies link the administration of cisplatin during pregnancy with fetal malformations [39]. Serkies et al. (2011) studied the effect of paclitaxel and cisplatin on the treatment of non-epithelial ovarian cancer during pregnancy. The authors found that there was no teratogenic fetal effect from the drugs [40]. Furthermore, etosopide appeared to have no effect on pregnancy if administered after the first trimester. On the other hand, a case of fetal exposure to bleomycin and etoposide in the period of 25 to 28 weeks of gestation was reported, in which cerebral atrophy was caused [41].

### 2.5. Pregnancy and Anti-Cancer Drug Treatment for Thyroid Cancer

A proportion of 10% of thyroid cancer is diagnosed during gestation. This type of cancer is second in its frequency during gestation [42].

The treatment of choice in thyroid cancer is surgery (total or partial resection of the gland) (in combination or not with radiotherapy), which is usually postponed until the end of childbirth unless the course of the disease is rapid and aggressive [43]. The oncological-pharmacological treatment approach for this cancer is a rare choice, and it is usually carried out in conjunction with radiotherapy. The oncological pharmaceutical compositions used mostly to address bone thyroid cancer and anaplastic thyroid cancer are the following: dacarbazine (cytotoxic agent, contraindicated strictly in pregnancy cases), vincristine (charged for embryotoxicity or teratogenic), cyclophosphamide (charged for embryonic teratogenesis, and carcinogenicity), streptozoic, doxorubicin, fluoroulacil, paclitaxlel, docetaxel, and carboplatin. The oncological pharmacological approach to thyroid cancer is avoided during pregnancy [44,45,46].

### 2.6. Pregnancy and Anti-Cancer Drug Treatment of Lymphomas (Figure 5)

According to Pereg and colleagues (2007), lymphoma is the fourth most common form of cancer diagnosed during pregnancy. Radiotherapy and chemotherapy in the first three months of the pregnancy have been associated with high rates of fetal congenital malformations. Therefore, these are strictly contraindicated. Upon completion of the first three months of gestation, and as time passes, the risk appears to be tempered [47]. The most widespread lymphoma diagnosed during pregnancy is lymphoma Hodgkin, and the most frequent chemotherapeutic approach comes in the form of doxorubicin–bleomycin–vinblastine and dacarbazine, or otherwise, ABVD [48,49].

**Figure 5 pharmaceutics-14-02080-f005:**
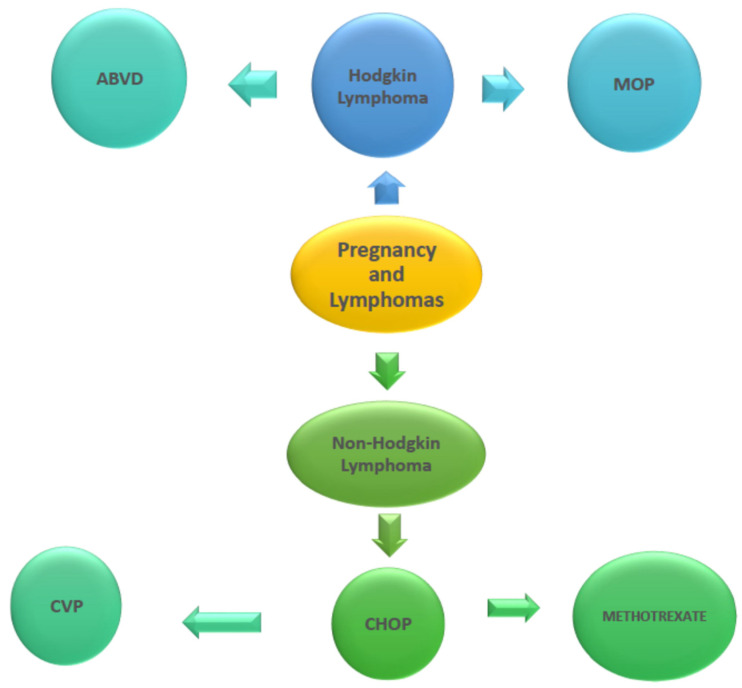
Pregnancy and lymphomas. ABVD: doxorubicin, bleomycin, vinblastine, dacarbazine; MOP: mechloretamine, vincristine, procarbazine, prednisone; CVP: cyclophosphamide, vincristine and prednisone; CHOP: cyclophosphamide, vincristine, prednisone, and adriamycin.

According to a case study, this chemotherapy regimen was administered during a twin pregnancy at 28 weeks and 3 days. Two cycles of treatment were administered at an interval of about two weeks. The male infant developed postpartum left heart malfunction, which was restored after four weeks. On the other hand, the female infant did not develop a heart malfunction [50]. A larger study revealed the effects of lymphoma chemotherapy (ABVD and MOPP, i.e., mechloretamine, vincristine, procarbazine, and prednisone). They were studied during pregnancy in a total of 84 offspring. The results of the study proved the safety of this chemotherapy for lymphomas during pregnancy. The children were all healthy after a long run of 29 years [51].

Non-Hodgkin lymphomas have not been shown to be particularly common during a pregnancy [49]. When lymphoma is still in its infancy, and the patient remains asymptomatic, it is recommended that the treatment approach be postponed until after delivery [52]. On the other hand, when there is an advanced stage lymphoma, the administration of treatment regimens is deemed necessary, and some of the following are selected:Cyclophosphamide, vincristine, and prednisone (CVP): This therapeutic option does not appear to present significant defects in fetuses if administered after the first three months of gestation [52]. However, it is reported that it can adversely affect the fertility of people of a reproductive age. At the same time, the pregnant woman receiving treatment may experience the following: neutropenia, thrombocytopenia, anemia, alopecia, insomnia, fatigue, peripheral neuropathy, etc. Therefore, the treating physician should be suspicious and closely monitor the progress of the patient [53];Cyclophosphamide, vincristine, prednisone, and adriamycin (CHOP): This scheme has proven to be safe if administered after the first trimester but not earlier [52];Methotrexate in high doses: This approach is chosen when lymphoma is at an advanced stage, but unfortunately, methotrexate administration during pregnancy is a safe option for the pregnant women, yet it is usually avoided due to increased adverse effects during pregnancy [54,55].

### 2.7. Pregnancy and Other Cancers

The anti-cancer drug approaches for the most common cancers during pregnancy were discussed above. What follows is a report on the less common but equally important cancers which may occur and complicate pregnancy.

### 2.8. Pregnancy and Lung Cancer

This form of cancer does not often occur during pregnancy, and the number of studies on this subject is extremely limited [13,56]. Statistics confirm that half of these cases are positively related to a medical history of smoking. The NSCLC (non-small cell lung cancer)- type adenocarcinoma is the most common form of lung cancer, which, indeed, correspond to 80% of lung cancer cases during gestation [57]. There are some chemotherapeutic regimens that can be safely administered to pregnant women after the end of the first trimester of pregnancy [13,35,56]. The schemes that are mostly used include cisplatin or carboplatin in combination with paclitaxel, docetaxel, vinorelbine, gemcitabine, or etoposide [13,56]. The effects on fetuses are transient respiratory distress and congenital malformations, and both of these are related to the administration of the regimens during the first three months of pregnancy [58].

### 2.9. Pregnancy and Brain Tumors (Figure 6)

Due to the rarity of this type of cancer in pregnant women, not many cases have been studied [13]. However, this condition, due to its specificity, continues to be a major clinical challenge [59].

**Figure 6 pharmaceutics-14-02080-f006:**
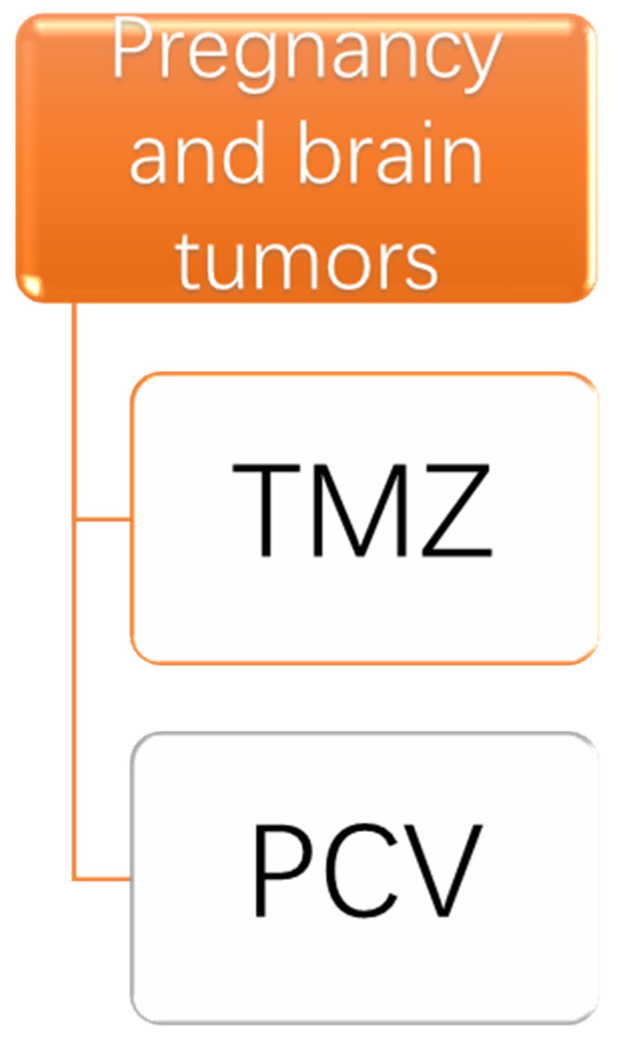
Pregnancy and brain tumors. TMZ: Temozolomide; PCV: Procarbazine, Lomustine and Vicristin.

In California, Blumenthal and colleagues (2008) studied six cases of brain glioma, which were treated when found to coexist with gestation status. Patients received oncology medication during the first month of pregnancy which occurred unintentionally. All six patients chose to stop chemotherapy and continue their pregnancies. Half of the cases received PCV (procarbazine, CCNU [lomustine], and vincristine, and the other half were treated with temozolomide (TMZ). The women did not appear to have any complications during pregnancy, and the infants were perfectly healthy in the long run, up to one year after delivery [59]. Additionally, a patient suffering from regenerative oligodendroglioma, receiving temozolomide (TMZ) therapy, immediately paused oncology treatment when she was found to be pregnant. She gave birth to a perfectly healthy infant without prenatal or perinatal complications [60].

### 2.10. Pregnancy and Leukemia (Figure 7)

Leukemia occurs in an amount of about one in a thousand pregnant women, and mostly in acute and especially medullary forms [61]. Regarding chemotherapy drugs, they should be avoided during the first three months of pregnancy and weeks before parturition [61]. Regarding chronic leukemia, the most common chemotherapy drug is imatinib (tyrosine inhibitor–kinase). How closely the administration of this is related to the appearance of fetal congenital malformations and spontaneous abortion is another controversial issue [28]. There have been observed infants who were perfectly normal and infants with serious and complex teratogenesis [62]. Similar results have been found in pregnancy studies after the administration of another tyrosine-kinase inhibitor, nilotinib [63].

**Figure 7 pharmaceutics-14-02080-f007:**
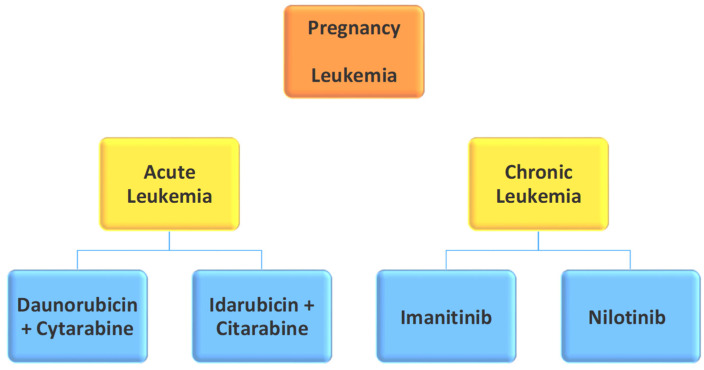
Pregnancy and Leukemia.

Acute leukemia is an aggressive form of cancer, and chemotherapy should be started as soon as possible; time delays are considered unacceptable. Acute myeloid leukemia (AML) is the most common leukemia diagnosed during pregnancy because it represents more than two-thirds of leukemia cases during pregnancy and has a frequency of 1 in 75,000–100,000 [49,64]. It should be organized within an oncology drug plan, which is separate for each clinical situation, while normally, the administration of daunorubicin or idarubicin in combination with cytarabine constitutes the first-line chemotherapy [64].

Chang and Patel (2015) conducted a systematic literature review aiming to determine the effects of chemotherapy against AML (daunorubicin or idarubicin with cytarabine) in pregnant women. The study showed that, when administering treatment after the first 3 months of pregnancy, the risk of fetal adverse effects decreases. The later the administration, the lower the risk tends to become. More specifically, during the first trimester, fetal deaths reached a rate of 37.5%, which was reduced to 9.7% in the second trimester, down to 0% in the third trimester [65].

### 2.11. Pregnancy and Acute Promyelocytic Leukemia

Acute promyelocytic leukemia (APL) is a type of acute myeloid leukemia (AML) with unique molecular pathogenesis, clinical manifestations, and therapeutic approaches. It is characterized by the proliferation of malignant promyelocytes in the bone marrow that carries a chromosomal shift or the PML/RARA gene (diagnostic criterion). Treating pregnant women with acute promyelocytic leukemia (APL) is an extremely difficult situation, where there is limited evidence. The studies found in the literature are limited, and usually concern only a few series of patients [66].

The oncological pharmaceutical disease approach constitutes mostly administering the all-trans retinoic acid (ATRA), frequently combined with other drugs such as daunorubicin or idarubicin, anthracycline, and arsenic trioxide (ATO) [51,67]. Analyzing the results of various studies, it appears that the administration of all-trans-retinoic acid during the last two trimesters of a pregnancy does not cause serious adverse maternal or fetal effects [49,67,68,69].

## 3. Cancer Pregnancy Treatment and Cardiotoxicity

The cardiotoxicity of cancer therapies refers to conditions that decline cardiac functioning by three different mechanisms: (1) the direct damage caused to cardiomyocyte functioning (toxic cardiomyopathy) termed as ‘’type I cancer therapy-related cardiomyopathy’’, (2) indirect damage caused to cardiomyocyte functioning via alterations related to the innervation, perfusion, or hormonal milieu of cardiomyocytes, termed as secondary ‘’type II cardiomyopathy’’, and finally, (3) myocarditis or ‘’type III cancer-related cardiomyopathy’’ caused by the activation of the inflammatory cascade against myocardial cells. Cancer therapy usually involves more than 2 different mechanisms of cardiomyopathy toxicity, but the overall classification of the mechanisms inducing cardiac toxicity serves to provide the best possible anticancer treatment. Regardless of the mechanism by which cardiotoxicity is induced the most common symptoms include arrhythmias, atrial fibrillation, bradycardia, QTc prolongation, and VT [70].

Nowadays, cardiotoxicity in cancer treatment is very important, as new concepts in the field of cardio-oncology emerge (by the use of immunotherapy and targeted molecular therapy), in addition to the concomitant prevalence of both cardiovascular diseases and cancer, and the overall aging population. Patients undergoing oncological treatment, as well as pregnant patients, should be evaluated during, before, and after the administration of anticancer regiments to evaluate their individual cardiovascular risk [70].

## 4. Immunotherapy during Pregnancy

Besides the surgical approach and chemo-, radio-, and hormonal therapy, immunotherapy has, in recent years, witnessed an expansion in its indications in the treatment of cancer. Coupled with the fact that even more women choose nowadays to postpone parenthood, thus increasing their chances of having some kind of malignancy during pregnancy, more and more are eligible for receiving immunotherapy during this period of their lives. Given the delicate balance between protecting against infections, as well as abnormal cells, and not rejecting the semi-allogenic fetus that the immune system of the mother tries to maintain, combined with the little knowledge and experience regarding the safety of this kind of medication for the fetus and the neonate, it is of vital importance to investigate whether immune-modifying agents could lead to unfavorable complications during pregnancy or lactation, and whether they are effective in treating the underlying cause during pregnancy and whether they could affect fertility [71,72,73].

A complicated state in a woman’s life is the maternal immune system during pregnancy. It deploys tolerance to the semi-allogeneic fetus, expressing both maternal and paternal antigens. At the same time, it endures the ability to protect against infections and toxins. Three immunological phases are distinct in this procedure: the first one is a pro-inflammatory state in the early stages of pregnancy, which is important for implantation and placentation in the uterus; the second one is found in the second and third trimesters, where an anti-inflammatory environment is developed in the uterus by cell populations, and the last part is associated with parturition. In this stage, uterine contractions, the delivery of the fetus, and the expulsion of the placenta are initiated by a second pro-inflammatory stage.

The mainstay of immunotherapy is the “immune-checkpoint inhibitors” (ICI), which include the anti-PD 1 (programmed cell Death), the anti-PD-L1 (programmed cell death ligand), and the anti-CTLA-4 (cytotoxic T-lymphocyte antigen) agents. Other agents include some cytokines, mainly IFNa (Interferon a) and IL-2 (Interleukin), vaccines, like BCG and a genetically engineered type 1 herpes simplex virus, specific T-cell engagers, and other immunomodulatory agents like pomalidomide. Some immune-modifying drugs, like the ICIs, are considered by the FDA to be category D drugs, while others, like the derivatives of thalidomide, are considered teratogenic and, thus, category X [74].

Regarding the immune-checkpoint inhibitors (ICIs), clinical data from seven women (six being treated for melanoma and one for trophoblastic tumor) who either conceived during therapy or initiated therapy during pregnancy, show that, while first-trimester exposure didn’t result in congenital malformations, second- and third- trimester exposure did lead to intrauterine growth restriction [75,76,77,78,79,80,81]. Four women gave birth prematurely; one neonate suffered from congenital hypothyroidism, which regressed spontaneously six months later, and another neonate was born without the left hand, a complication which was attributed most probably to mechanical factors and not to the therapy the mother received. [75,77,78,80]. Three mothers’ disease progressed, while the other four showed remission [76,77,78,79,80,81]. Furthermore, two patients experienced other complications, including diarrhea and liver-related toxicity, mainly jaundice, cholestasis, and elevated liver enzymes. As far as lactation is considered, although the breastmilk concentration of the drugs administered is lower than in the blood of the mothers, it is highly recommended that lactation be postponed for at least five months after the last dose. This is also a recommended period by the National Comprehensive Cancer Network before a woman should conceive.

As far as the rest of the agents are concerned, there is scarce data concerning their safety for the fetus and the neonate. Therefore, contraception is strongly advised during treatment and up to 30 days after the final dose, depending on the agent. The safest of the rest of the agents seems to be IFNa, probably due to the fact that it does not cross the placenta in substantial amounts [82,83]. It is also not found in significant concentrations in breast milk [84]. Again, due to the fact that data are collected from case reports, women are advised against conceiving during the treatment.

## 5. Cancer Treatment and the Role of Νutraceuticals

Nutraceuticals are either dietary supplements or herbal bioactive compounds, which have gained substantial attention in the past few years due to their potential therapeutic and nutritional advanced properties. They are currently used to prevent diseases and support optimal human body functioning, as well as improve quality of life and promote longevity. These bioactive compounds include lipids, alkaloids, carbohydrates, medicinal plants, edible flowers, etc. New research in the field of nutraceuticals shows their prominent beneficial results in the prevention of various diseases, including cancer, as well as preventing cardiotoxicity caused by various anticancer agents that are currently used in everyday practice. Typical examples include (1) carotenoids, including lycopene found in tomatoes, with known antioxidant properties, showing anticancer effects by reducing DNA damage and oxidative stress, (2) pectin, which impedes metastasis, (3) and phenolic compounds, including curcumin, caffeic, ferulic, and gallic acids, with known anticarcinogenic, anti-inflammatory, and antioxidant properties [71].

Moreover, very promising research in the field of biodegradable nanocarriers as useful pharmacological tools in clinical practice (to increase cellular and tissue accumulation of chemo-sensitizing agents and/or drugs in cancer therapy) provides new interesting therapeutic opportunities for cancer therapy. A very interesting example in the field of nanocarriers usage for anticancer treatment is the research of Barbarisi et al., which investigated the use of hyaluronic acid nanohydrogel loaded with the flavonoid quercetin, in combination with temozolomide, for glioblastoma tumor treatment. The nanohydrogel of quercetin seems to promote the internalization and enhancement of the temozolomide action in glioblastoma cells. Further research, of course, is necessary to investigate even more of the molecular mechanisms involved both in vitro and in vivo, regarding the biodistribution in vital organs, the pharmacokinetic and safety profile of the nanocarriers, along with different pharmacological agents [72].

## 6. Conclusions

The oncological treatment of pregnant women with cancer is not an easy task. Ensuring the good health of the patient is the primary concern; however, an effort should always be made to ensure the smooth outcome of the pregnancy. Unfortunately, many drugs cannot be used safely during pregnancy. Therefore, physicians should thoroughly examine all of the possible risks and complications from these systemic treatments before they make any treatment decision. In other words, the situation is extremely complex, but a few studies have been carried out regarding this issue. The majority of these studies conclude that, oncological drug therapy during the first three months of a pregnancy should be avoided as far as possible because it can cause fetal death, fetal teratogenic (and other) serious complications. The safest period for the granting of these treatments is after the 14th week of pregnancy.

Concerning immunotherapy, there are few data indicating its success. Further research is needed in order to reveal its efficacy during pregnancy. Data must be extracted and analyzed dutifully by physicians and, more specifically, by oncologists to be sure of the application of immunotherapy to pregnant women.

Finally, each patient case is unique and, in this way, should be treated by doctors. Each patient also requires an individualized approach. It is important before any treatment that all individual factors be weighed. Further studies are needed on the optimal oncological drug treatment for the various types of cancer diagnosed in pregnant women.

## Figures and Tables

**Figure 1 pharmaceutics-14-02080-f001:**
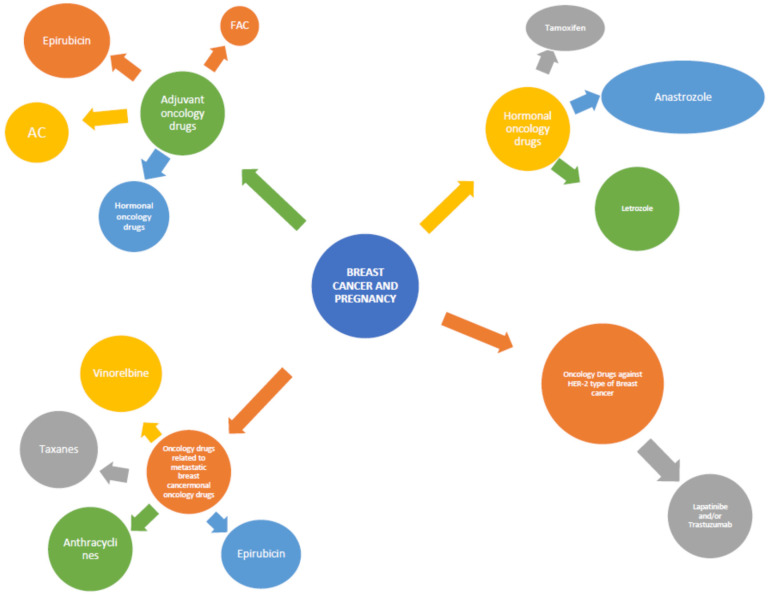
Breast cancer and pregnancy FAC: Fluoroucacil, Adriamycin, Cytoxan; AC: Doxorubicin, Cyclophosphamide.

**Table 1 pharmaceutics-14-02080-t001:** Treatment options during pregnancy according to cancer type.

Cancer Tissue	Therapeutic Choice					
**Breast Cancer**	** *(Adjuvant drugs)* **	**FAC**			**Epirubicin**	**AC**
	*(Hormonal treatment)*	Anastrozole			Letrozole	Tamoxifen
	*(Anti Her 2)*	Lapatinib			Trastuzumab	
	*(Metastatic cancer)*	Epirubicin		Taxanes	Vinorelbine	Anthracyclines
**Melanoma**		Cisplatin	Imatinib	Dabrafenib	Ipilimumab	Decarbazine + Bleomycine, Vincristine, Lomustine
**Cervical Cancer**		Cisplatin		Paclitaxel		Carboplatin
**Ovarian Cancer**		Taxanes (Paclitaxele/Docetaxele)			Platinum (Cisplatin/Carboplatin)	
**Lymphomas**	*(Hodgkin)*	MOPP			ABVD	
	*(Non-Hodgkin)*	Methotrexate	CVP			CHOP
**Brain Tumors**		Temozolomide			PCV	
**Leukemia**	*Acute*	Daunorubicin			Cytarabine	Idarubicin
	*Chronic*	Imatinib			Nilotinib	

FAC: Fluoroucacil + Adriamycin+Cytoxan; AC: Doxorubicin, Cyclophosphamide; MOPP: mechloretamine, vincristine, procarbazine, prednisone; ABVD: doxorubicin, bleomycin, vinblastine, dacarbazine; CHOP: Cyclophosphamide, vincristine, prednisone, Adriamycin; CVP: Cyclophosphamide, vincristine, prednisone; PCV: procarbazine, CCNU (lomustine), vincristine.

## Data Availability

Not Applicable.

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
