# Peer review of "Cancer Treatment and Immunotherapy during Pregnancy"

_pharmaceutics, 2022, doi:10.3390/pharmaceutics14102080_

Round 1

Reviewer 1 Report

This study is comprehensive and very detailed. The study data are of interest and valuable. The manuscript is very well written without larger mistakes. The minor observations are indicated below. My suggestion is to accept it with minor changes.

Introduction:

The subtitle in this part of the text should be avoided. References are not arranged well from the beginning of the Introduction.

Cytarabine is two times differently written in Figure 7.

Author Response

Dear Reviewer,

Thank you for your time and kind comments on reviewing our manuscript. Considering your suggestions, we proceeded on correcting the numeration of the References on the whole manuscript. Moreover, we rearranged the titles in order to avoid the use of a subtitle in the introduction section. The cytarabine is now corrected while minor grammar changes have been corrected through the whole manuscript as well.

Sincerely yours,

Koutras Antonios

Reviewer 2 Report

In this article, a wide range of antineoplastic treatments in pregnancy have been reported. The information is useful at the moment although it might have the risk of becoming old quite soon as this is a hot topic in research and new treatments might be developed anytime soon. However, it is still an interesting report as there is an ethical concern about research and pregnancy, so this can be useful for patients, obstetricians and gynaecologists. The two main things I believe need to be corrected before accepting this article for publication are:

1.       The text is full of some words, normally drug names, which I believe come from Greek (sisplatinis, paklitaxele) but are not acceptable in a review written in English, please check and correct.

2.       The article shows 7 figures which are basically the same thing but for different types of cancer. I think all the information can be summarised in just one figure or table.

Minor comments:

·         Line 66: please change the verb as ‘are born’ is not correct. Another alternative verbs could be: has appeared, has been raised…

·         Line 72: please rephrase.

·         Line 117: Figure 1 should be reported through the text rather than in the title.

·         Line 123: what is epirouvykini? Is that a brand name for epirubicin?

·         Figure 1: please add to the legend to meaning of abbreviations (FAC, AC). Different colours could be utilised in this figure, using yellow or light grey with white font makes it difficult to read.

·         Line 166: damprafenimpi?

·         Line 210: please rephrase, the adverse effects are the one appearing, not the patient ant the fetus.

·         Line 211: cisplatin, not sisplatinis.

·         Figure 4: paclitaxel instead of paklitaxele and docetaxel instead of docetaxele.

·         Line 285: please report what NSCLC means.

Author Response

  1. Drug names are now correct throughout the text.
  2. All figures are summarized in one table while in all figures the abbreviations are expanded in each legend. 

Considering minor changes:

  • Line 66 is now changed to "Thus, the so-called ethical dilemmas about starting the administration of oncological drugs to pregnant women has been raised"
  • Line 72 is being rephrased in order to be comprehensive.
  • Line 117: Figure 1 is now reported through the text.
  • Line 123: Epirouvykini is changed to epirubicin.
  • Figure 1: abbreviations are mentioned in the legend and the colours are changed to be more readable.
  • Line 166: damprafenimpi was written by mistake and is now changed to Dabrafenib
  • Line 210: The paragraph is rephrased.
  • Line 211: cisplatin is corrected.
  • Figure 4: paclitaxel and docetaxel are now corrected.
  • Line 285: NSCLC abbreviation is explained.